# FINE-GRAINED SEPARATION OF ACTION-BACKGROUND FOR POINT-LEVEL TEMPORAL ACTION LOCALIZATION

## ABSTRACT

Due to the limitation of coarse-grained video-level labels, the action-background confusion is a tough problem for the weakly-supervised temporal action localization. Point-level temporal action localization recently utilizes point-level labels to overcome this difficulty to some extent. However, considering the sparsity of point-level labels, existing methods still lack the ability to effectively eliminate false positive action proposals. To address this issue, in this paper, we propose a new framework to provide guidance for fine-grained separation of action-background for the model. Specifically, the framework relies on annotated single frame labels to extend the original action features and generate dense pseudo labels, providing the model with more precise position information. Based on this information, the framework generates pseudo segment-level labels from video sequences and utilizes our proposed score contrast module and feature separation module, which are different from the previous works,to amplify the differences in scores and features between segment labels. Extensive experiments on four benchmarks verify the effectiveness of our proposed framework, and demonstrate that our method is significantly superior to previous state-of-the-art methods and obtains 3.9% performance gains in terms of the average mAP on THUMOS'14.

## 1 INTRODUCTION

Temporal action localization is a crucial task in the video understanding field that involves identifying the start and end timestamps of various actions in untrimmed videos (Rashid et al., 2020; Ma et al., 2005; Xiong et al., 2019), while concurrently predicting their respective categories. Although numerous existing works have accomplished remarkable performance under the fully-supervised setting (*i.e.*, frame-level labels) (Zhang et al., 2019; Zhao et al., 2020; Zeng et al., 2019; Shou et al., 2017; Xu et al., 2020), the extremely expensive cost associated with obtaining such labels has led researchers to devise weakly-supervised methods (Wang et al., 2017; Shi et al., 2020; Liu et al., 2019; He et al., 2022). These methods only need video-level annotations for action categories, which is simpler to gather and more efficient for creating extensive datasets.

Typically, most weakly-supervised methods follow a hypothesis that video segments providing greater support to video-level classification are more likely to be action. However, this is often not the case. On the one hand, the lack of explicit location guidance makes it difficult for the model to determine the position of the action and background. On the other hand, several background segments that are more relevant to the video-level classification would further mislead the model into a dilemma of action-background confusion.

To address these difficulties and improve the performance while maintaining lower costs, point-supervised methods (Moltisanti et al., 2019; Ju et al., 2020) have begun to attract attention amongst researchers. In this setting, point-level labels provide approximate localization and quantity of action instances for the model, and the cost of point-level label is quite similar to that of the video-level label and significantly cheaper than frame-level label (45s vs. 50s vs. 300s per minute of video) (Ma et al., 2020). While recent works (Lee & Byun, 2021; Yang et al., 2022; Jing Tan, 2022) can apply point-level annotations that provide more information, they are inevitably subject to a large solution space, leading to high false-positive rates and generating discontinuous actions, which are

similar to many weakly-supervised frameworks. We employ the diagnostic tool (Alwassel et al., 2018) to perform the error analysis of BackTAL (Yang et al., 2022) and ASM (He et al., 2022), as shown in Figure 1. As expected, due to the use of point-level labels, BackTAL performs slightly better than the advanced weakly-supervised method ASM. However, from the diagnosing results, the performance of both are not excellent. For these two methods, the vast majority of errors come from the Localization Err. and the Background Err. among five types of errors. Obviously, the high error rates indicate the problem that the action and the background are difficult to be distinguished and separated is still not well solved.

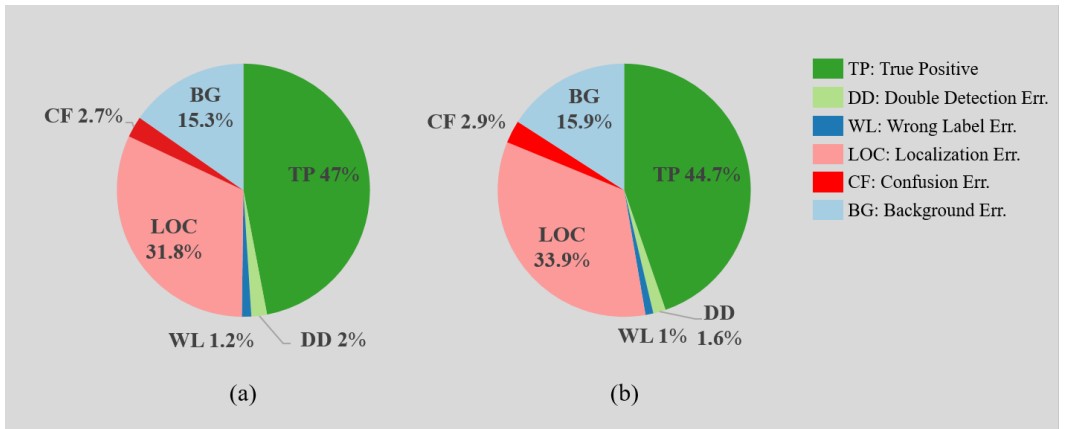

Figure 1: The performance analysis of Top 1 Prediction at IoU=0.5 on THUMOS14 for two methods. Illustration of these five types of errors are defined in detail in Alwassel et al. (2018). It is worth noting that other four types of errors except DD are greatly affected by action-background confusion. (a) The diagnosing results of BackTAL. (b) The diagnosing results of ASM.

How to effectively separate actions and backgrounds to improve the recognition performance and reduce error rates has become the primary research topic. In this paper, we propose a novel learning framework to achieve the fine-grained separation of action-background under the point-supervised setting called FS-PTAL. This fine-grained separation is considered from two perspectives: firstly, the intensity of the position information of the action and the background is enhanced; secondly, amplify the differences between the action and the background. The overall architecture of our approach is illustrated in Figure 2. Technically, we first design a label extension module and optimize a pseudo label mining strategy to strengthen the position information of the actions and backgrounds in original video sequence. Secondly, we rely on the precise position information which are provided by point-level labels to create a distribution copy of action-background aligned with the video features, and to generate pseudo segment-level labels that contain relatively complete instances of actions or backgrounds. In addition, to respectively intensify the score differences and the feature similarity differences between actions and backgrounds, we optimize the calculation of OIC (out-inner-contrastive loss) and redesign the score contrast module, and introduce feature embedding space and point-level cosine similarity separation in the feature separation module. Finally, We re-design four losses to better separate the action and the background and improve performances in the detection and localization.

In summary, our contributions are as follows:

1) We design a novel framework to achieve fine-grained separation of action-background under the point-supervised setting. Compared with the previous methods, it can effectively overcome the difficulty of action-background confusion but with similar annotation costs.

2) We utilize the label extension and mining strategy to enhance the location information of the action and the background, and optimize a score contrast module and propose a novel feature separation module to magnify the difference of action-background. On this basis, we redesign the composition of the video-level loss, the point-level loss and the score contrast loss, and introduce the feature separation loss.

3) Extensive experiments are performed on four benchmarks to prove the effectiveness of each module. Experimental results show that FS-PTAL achieves a new state-of-the-art on these datasets, e.g., 56.74% with the average mAP on THUMOS'14.

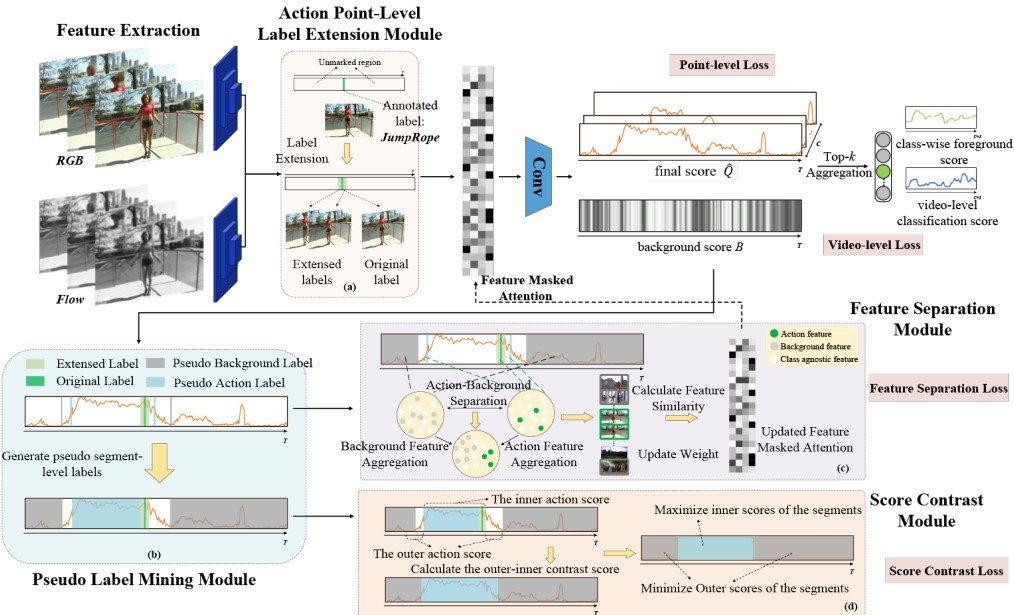

Figure 2: Framework of the proposed FS-PTAL. We extract video features and classify each frame to obtain the final score $\widehat{Q}$. This framework contains four important modules: (a) Label extension module (see Sec. 3.1). (b) Pseudo label mining module (see Sec. 3.3). (c) Feature Separation Module (see Sec. 3.5). (d) Score contrast module (see Sec. 3.4).

## 2 RELATED WORK

**Temporal action localization(TAL)**. TAL methods rely on precise annotations for each frame. These methods can be categorized into one-stage and two-stage approaches. The two-stage paradigm (Dai et al., 2017; Sridhar et al., 2021; Zhu et al., 2021) firstly generates action proposals, and then classifies the proposals and refines the boundary. The majority of proposal generation methods rely on anchor mechanisms (Chao et al., 2018; Xu et al., 2017; Yang et al., 2020), but there are also other ways to generate proposals, including sliding window approaches (Shou et al., 2016) and combining confident starting and ending frames (Lin et al., 2019) which are classified by the temporal action grouping method. Many recent works focus on the one-stage methods (Lin et al., 2021; Long et al., 2019; Xu et al., 2020) that show excellent performance while having a relatively simple structure, which predict the boundaries and labels of actions. ReAct (Shi et al., 2022) proposes a DETR-like Carion et al. (2020) framework to approach temporal action localization. Zhao et al. (2023) designed the Movement Enhance Module to explore local and temporal relations between snippets. Kim et al. (2023) utilized cross-attention maps to provide feedback to self-attention of the encoder and decoder.

**Weakly-supervised temporal action localization(WS-TAL)**. Weakly-supervised methods aim to address the high cost of annotation by relying on video-level category labels. There are two main branches of methods: MIL-based (Lee et al., 2020; Luo et al., 2020; Ma et al., 2021; Moniruzzaman et al., 2020) and attention-based frameworks (Hong et al., 2021; Liu et al., 2022; Luo et al., 2021; Narayan et al., 2021). Recently, there have been a few new Weakly-supervised researches. ASM-Loc (He et al., 2022) introduced three novel segment-centric modules for action-aware segment modeling beyond standard MIL-based methods. RSKP (Huang et al., 2022) identified the representative snippets in each video to propagate information between these snippets, so as to generate better pseudo labels. Ju et al. (2023) extracted unrestricted action information from readily available VLP models to facilitate WTAL. Liu et al. (2023) proposed a clustering-based F&B separation algorithm

for WTAL. Zhou et al. (2023) came up with an effective pipeline for learning better pseudo labels. Tang et al. (2023) introduced a new graphical network for modeling ambiguous and discriminative segments with different connection types. However, due to their reliance on insufficient labels and the need for empirical preset thresholds, these methods often encounter difficulties of background false positives and incomplete action predictions. Therefore, there is a significant performance gap between weakly-supervised methods and fully-supervised methods.

**Point-Level supervised temporal action localization**. It is a new kind of weakly-supervised paradigm for temporal action localization, which is widely used to balance labeling costs and model performance. Moltisanti et al. (2019) firstly annotated the single timestamp of each action in place of expensive action boundaries. SF-Net (Ma et al., 2020) utilized the pseudo label mining strategy to acquire more labeled frames, obtaining certain performance improvements. Ju et al. (2020) introduced the proposal-based prediction paradigm for point-level labels. BackTAL (Yang et al., 2022) proposed background-click supervision to replace the previous action-click supervision to reduce background error rates. LACP (Lee & Byun, 2021) generated the dense optimal sequence to provide completeness guidance for the model. Although the performance of the above work is better than that of the weakly-supervised method, there is a significant performance gap compared to fully-supervised methods due to the interference of action-background confusion. Our proposed FS-PTAL, which extends action point-level annotations and mines pseudo action and background point-level labels to enhance the location information of the instances. At the same time, we use the score contrast module and the feature separation module to intensify the difference between the action segments and the background segments, and provide the guidance of fine-grained separation of action-background for the model. In Sec. 4, the effectiveness of our method is clearly verified through numerous experiments.

## 3 OUR APPROACH

In this section, as shown in Figure 2, we first describe the problem definition and the feature extraction. Afterward, we demonstrate the action point-level label extension module and the baseline setup. Then, we provide a detailed explanation of the pseudo label mining strategy and the related point-level loss. Finally, the score contrast module and the feature separation module are elaborated exhaustively.

**Problem definition**. The action point-level label of an input video is defined as $P^{act} = \{t_i, p_{t_i}\}_{i=1}^{N^{act}}$, which is processed in Sec. 3.1, here the $i$-th action instance is marked with its action label $p_{t_i}$ at the $t_i$-th frame, and $N^{act}$ is the total number of action instances for this input video. We use the binary vector label, denoted by $p_{t_i}$, where $p_{t_i}[k] = 1$ if the $i$-th action instance contains the $k$-th action class and 0 otherwise. These point-level labels can be aggregated to obtain video-level labels, denoted by $y^{vid} \in \mathbb{R}^C$. During inference, we generate a set of action proposals $\widetilde{S} = \{s_n, e_n, c_n, q_n\}_{n=1}^{N^{pro}}$ with a quantity of $N^{pro}$, where each proposal is represented by the predicted action class $c_n$, the confidence score $q_n$, and start and end times $s_n$ and $e_n$.

**Feature extraction**. To process a video, we first divide it into 16-frame segments and input them into a pre-trained feature extractor that has been trained on the Kinetics-400 dataset. Following Liu et al. (2019), the RGB and flow stream features are combined by concatenation to create $X \in \mathbb{R}^{T \times D}$, where $T$ represents the number of segments and $D$ represents the feature dimension. It is worth noting that the $X$ here is the data feature that has been extended in Sec. 3.1. Next, the extracted features are passed through a convolutional layer followed by ReLU activation, resulting in embedded features $F \in \mathbb{R}^{T \times D}$. We utilize $F$ as the input for our model.

### 3.1 LABEL EXTENSION MODULE

As shown in Figure 2(a), in a long video sequence, the labeled action point-level annotations only occupy a small interval, and the surrounding large area is all unlabeled. The aim of this module is to extend the information coverage of action point-level annotations and enhance the strength of position information. In He et al. (2022), to compensate for the insufficient information of actions within a short duration, dynamic segment sampling was designed. However, this method judges the position information of actions with short duration based on the predicted action proposals in each round, which intuitively relies heavily on the accuracy of the action proposals and has quite

limited performance improvements. In contrast, based on the precise position information provided by point-level labels, we propose a label extension module to increase the number of action point-level annotations and the length of the feature.

Firstly, we initialize the sampling weight vector $W \in \mathbb{R}^{T_{ori}}$, with values equal to 1 at all time steps. Here, $T_{ori}$ is the feature length before extension. Secondly, given the original action point-level label $P_{ori}^{act}$, we obtain a list of label indexes $Idx_{label}$, and group consecutive indexes into a group to generate a list of the label group indexes $Idx_{seg} = \{seg_{sid}\}_{sid=1}^{N_{sid}}$, where $sid$ is the index of the current group and $N_{sid}$ is the total number of the groups. We set the higher sampling weights $\tau_1$ and $\tau_2$ for the position and the surrounding of the labels respectively, which aims to increase the sampling proportion of these areas. For each $seg_{sid}$, we update the sampling weight as in formula (1).

$$W[seg_{sid}[0] : seg_{sid}[-1] + 1] = \tau_1$$
$$W[seg_{sid}[0] - 1] = \tau_2, W[seg_{sid}[-1] + 1] = \tau_2 \tag{1}$$

Afterwards, we compute the cumulative distribution function (CDF) of the sampling weights $f_{\mathcal{W}} = CDF(\mathcal{W})$, then sample $T$ time steps uniformly from the inverse of the CDF: $\{x_i = f_{\mathcal{W}}^{-1}(i)\}_{i=1}^{T}$. we apply Linear Interpolation to the original data $X_{ori} \in \mathbb{R}^{T_{ori} \times D}$, generating the data $X \in \mathbb{R}^{T \times D}$. With the feature length increasing, the positions of the original labels $P_{ori}^{act}$ must also shift and increase accordingly. The complete algorithm which displays the extension process will be presented in the Sec. A of **appendix**, and we are still in the **appendix** for a brief analysis of the formula (1).

## 3.2 BASELINE SETUP

Given the embedded features $F$, we apply a convolutional layer to predict the temporal class activation maps $\mathcal{M} \in \mathbb{R}^{T \times (C+1)}$ and intercept it to derive segment-level class scores $Q \in \mathbb{R}^{T \times C}$ and the class-agnostic background scores $B \in \mathbb{R}^T$. Referring to the fusion strategy of Xu et al. (2017), we acquire final scores $\widehat{Q} \in \mathbb{R}^{T \times C}$, *i.e.*, $\widehat{q}_t[c] = q_t[c](1 - b_t)$, here $b_t$ represents the background score. Afterwards, we utilize the top-k aggregation strategy to calculate the video-level classification score $Q_v \in \mathbb{R}^{C+1}$, and calculate the class-wise foreground confidences $Q_f \in \mathbb{R}^{C+1}$ as in Huang et al. (2021).

We concatenate the video-level labels $y^{vid}$ and value 0 to obtain the foreground class label $y_{fore}^{vid} \in \mathbb{R}^{C+1}$, and likewise concatenate it and value 1 to acquire the background class label $y_{back}^{vid} \in \mathbb{R}^{C+1}$. The video-level loss is composed of the foreground loss and the background loss. The foreground loss is calculated by applying the binary cross-entropy between $y_{fore}^{vid}$ and $Q_f$, written as formula (2). Similarly, the calculation of the background loss is presented in formula (3).

$$\mathcal{L}_{fore} = -\sum_{c=1}^{C} \left( y_{fore}^{vid}[c] \log Q_f[c] + (1 - y_{fore}^{vid}[c]) \log (1 - Q_f[c]) \right) \tag{2}$$

$$\mathcal{L}_{back} = -\sum_{c=1}^{C} \left( y_{back}^{vid}[c] \log Q_v[c] + (1 - y_{back}^{vid}[c]) \log (1 - Q_v[c]) \right) \tag{3}$$

The video-level loss of our approach is obtained by adding these two losses together, as shown in formula (4).

$$\mathcal{L}_{vid} = \mathcal{L}_{fore} + \mathcal{L}_{back} \tag{4}$$

## 3.3 PSEUDO LABEL MINING STRATEGY

Even with the extension of point-level labels, the sparsity of labels has not been solved, so it is necessary to mine pseudo labels containing actions and backgrounds in videos. The mining process is shown briefly in Figure 2(b). Specifically, considering that the actions and the backgrounds are often interspersed during the video, we firstly look for the segments whose background scores $b_t$ are larger than the threshold of the potential background $\gamma_{bkg}$ between two adjacent action labels. If not found, the segment with the largest background score will be selected for marking. Followed

by, the adjacent background segments can be concatenated into a longer one, enhancing the ability to discover the background from the environment with the action-background confusion.

Lee & Byun (2021) firstly proposes label mining strategy at point-level supervision. However, it is not effective to mine action labels. LACP is to search the segments whose background score $b_t$ are lower than the threshold of the potential action $\gamma_{act}$ as pseudo action labels between adjacent action points and background points, where $\gamma_{act}$ is set a quite low value to filter the background interference. Whereas, this also limits the ability to mine pseudo action labels. To optimize it, we raise $\gamma_{act}$ but add another limitation that the model must judge the mined action score of segment lower than $\gamma_{act}$ is whether the highest value in $\widehat{Q}$. If it matches, it is marked as a pseudo action label, otherwise skip it. The related algorithm and comparative experiments are shown in the Sec. B of **appendix**.

After the label mining algorithm, we will describe the original action point-level labels, *i.e.*, $P^{act} = \{t_i, p_{t_i}\}_{i=1}^{N^{act}}$, pseudo action point-level labels, *i.e.*, $P_{pse}^{act} = \{t_k, p_{t_k}\}_{k=1}^{N_{pse}^{act}}$, and pseudo background point-level labels, *i.e.*, $P^{bkg} = \{t_j\}_{j=1}^{N^{bkg}}$, respectively. The point-level loss is also calculated by the binary cross-entropy, which is composed of the classification loss for action points and background points. The calculation for the action point-level loss is shown as following formula (5), and $\mathcal{P} = P^{act} + P_{pse}^{act}, \mathcal{N} = N^{act} + N_{pse}^{act}$.

$$\mathcal{L}_{p_{act}} = -\frac{1}{\mathcal{N}} \sum_{(t,p_t)}^{\mathcal{P}} \sum_{c=1}^{C} \left( p_t[c](1 - \widehat{q}_t[c])^2 \log \widehat{q}_t[c] + (1 - p_t[c])\widehat{q}_t[c]^2 \log(1 - \widehat{q}_t[c]) \right) \quad (5)$$

The calculation for the background point-level loss is shown as following formula (6).

$$\mathcal{L}_{p_{bkg}} = -\frac{1}{N^{bkg}} \sum_{t}^{P^{bkg}} \sum_{c=1}^{C} \left( \widehat{q}_t[c]^2 \log(1 - \widehat{q}_t[c]) + (1 - b_t)^2 \log b_t \right) \quad (6)$$

At last, the point-level loss is defined as the sum of above two losses.

$$\mathcal{L}_{point} = \mathcal{L}_{p_{act}} + \mathcal{L}_{p_{bkg}} \quad (7)$$

### 3.4 SCORE CONTRAST MODULE

Based on the dense pseudo labels, we establish a distribution copy of action-background. Next, we splice pseudo point-level labels into pseudo segment-level labels. We extend the coverage of the segment-level labels by using the distribution copy and make these labels contain as complete actions or backgrounds as possible. Shou et al. (2018) firstly introduces OIC (out-inner-contrastive loss) to evaluate whether a segment contains complete instances, and this strategy has also been adopted by many works(He et al., 2022; Lee & Byun, 2021; Lee et al., 2018). However, there is a flaw in the general calculation (Lee & Byun, 2021; Shou et al., 2018) for the outer-inner contrast score. For example, in Figure 3(b), during a sequence with alternating long and short actions, the calculated scope of the outer score for a long action is very likely to contain the short action. The pseudo segment-level labels of the class-specific $c$ are defined as $SL_c = \{(s_n^c, e_n^c)\}_{n=1}^{N_{sl}^c}$, where $s_n^c$ and $e_n^c$ denote the start and end time for the label, $N_{sl}^c$ is the number of labels of class $c$. The calculated length $\delta l_{seg}$ of the outer score is only related to the segment length $l_{seg} = e_n^c - s_n^c + 1$. Obviously, this calculation is not what we really want. We optimize the calculation method for the outer scope as following formula 8), here $\delta$ is hyper-parameter adjusting the outer range. Based on previous works (Qu et al., 2021; He et al., 2022; Lee & Byun, 2021), we set $\delta$ to 0.25.

$$\begin{cases} left : (min(s_n^c - \delta l_{seg}, e_{n-1}^c), s_n^c - 1), if \ n = 1, \ e_{n-1}^c = 0 \\ right : (e_n^c + 1, min(e_n^c + \delta l_{seg}, s_{n+1}^c)), if \ n = N_{sl}^c, \ s_{n+1}^c = T \end{cases} \quad (8)$$

Then, we calculate the difference in $q_t$ between the inner and outer ranges to obtain $R(SL_c)$. At last, we maximize (set to 1) inner scores and minimize (set to 0) outer scores to acquire best outer-inner contrast score $R(SL_c^{best})$. The calculation of score contrast loss is as formula (9).

$$\mathcal{L}_{score} = \frac{1}{sum(y^{vid})} \sum_{c=1}^{C} y^{vid}[c](1 - R(SL_c^{best}))^2 \quad (9)$$

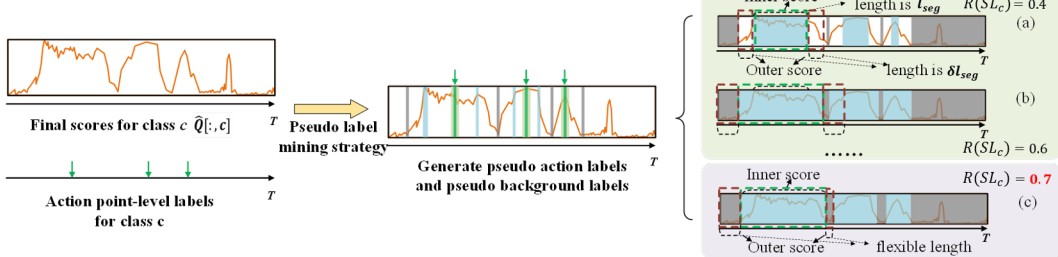

Figure 3: The process of outer-inner contrast score for class $c$. Given the final scores and the point-level labels, we mine dense pseudo labels. Then, We calculate the outer-inner contrast score among all possible candidates.

In the Sec. D of **appendix**, we compare the effects of pre-and post-optimization OIC calculations on experimental results.

### 3.5 FEATURE SEPARATION MODULE

Feature separation has been widely used by previous works (Min & Corso, 2020; Paul et al., 2018; Sun et al., 2020; Lee & Byun, 2021), and its aim is to guide the model easier for separating the actions from the backgrounds each other. We note that our module differs from Min & Corso (2020); Lee & Byun (2021) in that they use segment-level coarse-grained features and destroyed the integrity of feature information during the pooling and feature sampling, whereas ours introduces feature embedding space, which not only preserves complete feature information but also enhances it, we also contrast feature at point-level to achieve fine-grained separation. Due to the consideration of high space consumption and slow convergence speed, we abandon triplet loss, and due to the precise information provided by frame labels, the Contrastive loss also has a very excellent effect. Specifically, we firstly utilize two convolution layers to operate the input features to obtain a feature embedding space which distinguish actions from backgrounds and strengthen their feature information. Then, we calculate the cosine similarity between different frame sequences of the instances and obtain frame-specific attention weight, namely feature masked attention layer. As shown in Figure 2(c), this layer is added to the convolution calculation process to focus effectively on features of interest. Given the feature embedding space $E \in \mathbb{R}^{T \times D_{ebd}}$, $D_{ebd}$ is the embedding dimension of the single frame. Meanwhile, We define the cosine similarity between $e_1$ and $e_2$ by the following formula (10).

$$cos(e_1, e_2) = \frac{e_1 \cdot e_2^\top}{\|e_1\|_2 \cdot \|e_2\|_2} \tag{10}$$

Based on the action point-level labels $P^{act}$ and the pseudo background point-level labels $P^{bkg}$, we can get the action feature embedding $e_{act} \in \mathbb{R}^{T_{act} \times D_{ebd}}$ and the background feature embedding $e_{bkg} \in \mathbb{R}^{T_{bkg} \times D_{ebd}}$, where $T_{act}$ and $T_{bkg}$ are respectively the number of marked action frames and labeled background frames. Afterwards, the feature separation loss $\mathcal{L}_{fs}$ is calculated from three aspects, *i.e.*, between two background labels, between two action labels and between the action-background pair. Firstly, feature embedding vectors from two background frames should be similar to each other, and the loss $\mathcal{L}_{fs}^{bg}$ can be formulated as formula (11).

$$\mathcal{L}_{fs}^{bg} = max\left(\psi_{same} - \mathcal{H}\left(cos(e_{bkg}, e_{bkg})\right), 0\right) \tag{11}$$

Here, $\psi_{same}$ is the similarity threshold between frames from the same category. $\mathcal{H}()$ is the function to mine the hard example from the comparison of the feature embedding vector pair, which selects the minimum of the same category pair or the maximum of the different category pair from the second dimension. Similarly, the calculation of the remaining two aspects are as follows, and $\psi_{diff}$ is the threshold to constrain the similarity between actions and background.

$$\mathcal{L}_{fs}^{act} = max\left(\psi_{same} - \mathcal{H}\left(cos(e_{act}, e_{act})\right), 0\right) \tag{12}$$

$$\mathcal{L}_{fs}^{ab} = max\left(\mathcal{H}\left(cos(e_{act}, e_{bkg})\right) - \psi_{diff}, 0\right) \tag{13}$$

Table 1: State-of-the-art comparison on THUMOS14, including frame-level, video-level and point-level supervision methods. The average mAPs are calculated under the IoU thresholds 0.1:0.5 and 0.3:0.7 with the step 0.1.

| Supervision | Method | mAP@IoU(%) | | | | | | | AVG (0.1:0.5) | AVG (0.3:0.7) |
|---|---|---|---|---|---|---|---|---|---|---|
| | | 0.1 | 0.2 | 0.3 | 0.4 | 0.5 | 0.6 | 0.7 | | |
| Frame-level (Fully) | BMN (Lin et al., 2019) | - | - | 56.0 | 47.4 | 38.8 | 29.7 | 20.5 | - | 38.5 |
| | BC-GNN (Bai et al., 2020) | - | - | 57.1 | 49.1 | 40.4 | 31.2 | 23.1 | - | 40.2 |
| | BU-TAL (Zhao et al., 2020) | - | - | 53.9 | 50.7 | 45.4 | 38.0 | 28.5 | - | 43.3 |
| | AFSD (Lin et al., 2021) | - | - | 67.3 | 62.4 | 55.5 | 43.7 | 31.1 | - | 52.0 |
| | ReAct (Shi et al., 2022) | - | - | 69.2 | 65.0 | 57.1 | 47.2 | 35.6 | - | 55.0 |
| | MENet (Zhao et al., 2023) | - | - | 70.7 | 65.3 | 58.8 | 49.1 | 34.0 | - | 55.6 |
| | Self-DETR (Kim et al., 2023) | - | - | 74.6 | 69.5 | 60.0 | 47.6 | 31.8 | - | 56.7 |
| Video-level (Weakly) | FTCL (Gao et al., 2022) | 69.6 | 63.4 | 55.2 | 45.2 | 35.6 | 23.7 | 12.2 | 53.8 | 34.4 |
| | ASM-Loc (He et al., 2022) | 71.2 | 65.5 | 57.1 | 46.8 | 36.6 | 25.2 | 13.4 | 55.4 | 35.8 |
| | RSKP (Huang et al., 2022) | 71.3 | 65.3 | 55.8 | 47.5 | 38.2 | 25.4 | 12.5 | 55.6 | 35.9 |
| | DELU (Chen et al., 2022) | 71.5 | 66.2 | 56.5 | 47.7 | 40.5 | 27.2 | 15.3 | 56.5 | 37.4 |
| | Ju et al. (2023) | 73.5 | 68.8 | 61.5 | 53.8 | 42.0 | 29.4 | 16.8 | 60.0 | 40.8 |
| | Liu et al. (2023) | 72.3 | - | 59.2 | - | 37.7 | - | 13.7 | 57.1 | - |
| | DDG-Net (Tang et al., 2023) | 72.5 | 67.7 | 58.2 | 49.0 | 41.4 | 27.6 | 14.8 | 57.8 | 38.2 |
| Point-level (Weakly) | SF-Net (Ma et al., 2020) | 68.3 | 62.3 | 52.8 | 42.2 | 30.5 | 20.6 | 12.0 | 51.2 | 31.6 |
| | Ju et al. (2020) | 72.3 | 64.7 | 58.2 | 47.1 | 35.9 | 23.0 | 12.8 | 55.6 | 35.4 |
| | BackTAL (Yang et al., 2022) | - | - | 54.4 | 45.5 | 36.3 | 26.2 | 14.8 | - | 35.4 |
| | LACP (Lee & Byun, 2021) | 75.7 | 71.4 | 64.6 | 56.5 | 45.3 | 34.5 | 21.8 | 62.7 | 44.5 |
| | Ours | **79.1** | **74.6** | **68.6** | **61.0** | **50.7** | **38.9** | **24.2** | **66.8** | **48.7** |

At last, the feature separation loss jointly considers the above three terms and can be calculated as formula (14).

$$\mathcal{L}_{fs} = \mathcal{L}_{fs}^{bg} + \mathcal{L}_{fs}^{act} + \mathcal{L}_{fs}^{ab} \tag{14}$$

## 3.6 Joint Training

The overall training objectives of our model are as formula (15). Among them, $\lambda_*$ is the weighted parameter used to balance the loss, which are determined empirically. The inference will be explained in the Sec. C of **appendix**.

$$\mathcal{L}_{total} = \lambda_1 \mathcal{L}_{vid} + \lambda_2 \mathcal{L}_{point} + \lambda_3 \mathcal{L}_{score} + \lambda_4 \mathcal{L}_{fs} \tag{15}$$

## 4 Experiments

**Dataset**. We evaluate our FS-PTAL on four datasets: THUMOS14 (Jiang et al., 2014), ActivityNet v1.3 (Heilbron et al., 2015), BEOID (Damen et al., 2016) and GTEA (Lei & Todorovic, 2018). THUMOS14 contains untrimmed videos in 20 action class. THUMOS14 may has multiple instances of action in a single video with 200 and 210 videos (Wang et al., 2017; Zhai et al., 2020; Zhang et al., 2021) for validation and test, respectively. ActivityNet v1.3 is a large dataset containing 200 complex daily activities, which contains 10,024 training, 4,926 validation and 5,044 test videos. GTEA is a dataset containing 7 fine-grained daily actions in the kitchen, with 21 and 7 videos used for training and testing, respectively. BEOID consists of 58 videos with a total of 34 action categories.

**Experiment Setting**. Our method is implemented with Pytorch toolbox, and the overall network architecture is built on the GPU of Geforce RTX 3080. The parameter settings and experiments can be found in the Sec. C of **appendix**.

### 4.1 Comparison with the State of the Art

As shown in Table 1, our FS-PTAL is compared with state-of-the-art methods at different levels of supervision on THUMOS14. Obviously, our mthod is significantly superior to advanced WS-TAL

Table 2: Ablation experiments on THUMOS14. AVG represents the average mAP at the IoU thresholds 0.1:0.1:0.7. (*A*: label extension module, *B*: pseudo label mining module, *C*: score contrast module, *D*: feature separation module).

| *A* | *B* | *C* | *D* | mAP@IoU(%) | | | | | | | AVG |
|-----|-----|-----|-----|-----|-----|-----|-----|-----|-----|-----|-----|
| | | | | 0.1 | 0.2 | 0.3 | 0.4 | 0.5 | 0.6 | 0.7 | |
| ✗ | ✓ | ✓ | ✓ | 77.6 | 73.2 | 66.9 | 59.2 | 47.3 | 36.1 | 23.1 | 54.8 |
| ✓ | ✗ | ✓ | ✓ | 72.2 | 68.9 | 61.3 | 52.7 | 42.3 | 30.8 | 16.9 | 49.3 |
| ✓ | ✓ | ✗ | ✓ | 75.3 | 71.1 | 63.8 | 57.2 | 44.6 | 35.1 | 20.4 | 52.5 |
| ✓ | ✓ | ✓ | ✗ | 75.5 | 70.8 | 62.4 | 54.9 | 44.1 | 33.7 | 19.8 | 51.6 |
| ✓ | ✓ | ✓ | ✓ | **79.1** | **74.6** | **68.6** | **61.0** | **50.7** | **38.9** | **24.2** | **56.7** |

and point-level supervised methods. For example, both Avg(0.1:0.5) and Avg(0.3:0.7), ours are far better than that of LACP, with the improvement of nearly 4%. In particular, the performance gains at high IOU suggest that our method does indeed better separate the actions from the backgrounds. In addition, the performance gap between our approach and the fully-supervised methods has narrowed dramatically, and the performances at low IOU (*i.e.*, 0.3 or 0.4) are even better than that of some fully-supervised approaches. Due to the lack of action boundary information, the performance of our method at high IOU (*i.e.*, 0.6 or 0.7) still lags significantly behind fully-supervised methods.

Notably, We also provide extensive experiments which compare our FS-PTAL with state-of-the-art methods on ActivityNet v1.3, BEOID and GTEA. There is no doubt that our approach achieves the best performances on these three benchmarks. Considering the length of this paper, the experimental tables and the related analysis are in the Sec. D of **appendix**. Meanwhile, we also explore the impact of the use of different kinds of point-level labels (Ma et al., 2020; Moltisanti et al., 2019) on the final performance on THUMOS14 in the Sec. D.

## 4.2 ABLATION EXPERIMENT

We conducted four sets of ablation experiments to verify the effectiveness of our proposed module. Table 2 shows the results of ablation experiments. The label extension Module has the lowest performance gain 2.1% as it only extends point-level labels and does not affect the model framework. In contrast, due to the sparsity of labels, which may greatly affect the performance of the score contrast module and feature separation module, the performance gain brought by pseudo label mining module reaches its maximum value, which is 7.4%. In addition, the score contrast module and feature separation module mainly promote the separation of action-background from different perspectives, their performance gains are 4.2% and 5.1%, respectively. Finally, the significant performance improvement of the complete framework at all IOU thresholds clearly indicates that our method effectively provides the fine-grained separation guidance of action-background for the model.

## 4.3 QUALITATIVE COMPARISON AND VISUALIZATION

We provide qualitative comparisons with methods such as LACP and DELU, and the visualization results are presented in the Sec. E. In addition, we also employ the diagnostic tool (Alwassel et al., 2018) to compare and perform error analysis of our model with other SOTAs in the Sec. E. All analysis and comparison results indicate that our FS-PTAL has superior performance to other methods.

## 5 CONCLUSION

In this paper, we propose a novel point-supervised framework to guide the fine-grained separation of action-background. We utilize the label extension module and the pseudo label mining strategy to address the difficulty of label sparsity, and create a distribution copy of action-background based on dense pseudo labels. Next, we generate a large number of segment-level labels by aggregating point-level labels, and separate actions and backgrounds from two aspects by our proposed the score contrast module and the feature separation module. We perform a large number of experiments and the results show that our model achieves a new state-of-the-art with a large gap on four benchmarks.

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

## A    SUPPLEMENTARY INFORMATION FOR LABEL EXTENSION MODULE

Algorithm 1 shows the extension process of the label extension module in Sec. 3.1. The final point-level labels output by this algorithm will be used as input to our model, as shown in Sec. 3, for subsequent processing. In addition, for formula (1), we respectively set higher sampling weights $\tau_1$, $\tau_2$ for the position and surroundings of action point-level labels. This is because most action instances have a longer duration, and for them, the surroundings of action labels also contain action instances; But considering that for short actions, the surroundings of labels is more likely to contain backgrounds. Therefore, we set the weight value of $\tau_1$ to be higher than $\tau_2$ to reduce the negative impact of the surroundings that may contain background.

---

**Algorithm 1:** Extension on class-specific point-level labels

**input** : point-level labels of specific-class: $P^c_{ori} = \{(t_{ori}, p^c_{t_{ori}})\}_{t_{ori}=1}^{T_{ori}}$, feature array: $Sam \in \mathbb{R}^{T_{ori}}$, sampled feature array: $Sam_{etd} \in \mathbb{R}^{T}$, range hyper-parameter: $\vartheta$

**output:** extended point-level labels: $P^c = \{(t, p^c_t)\}_{t=1}^{T}$

1 **for** $(t_{ori}, p^c_{t_{ori}})$ *in* $P^c_{ori}$ **do**
2     $pf_{idx} \leftarrow Sam[t_{ori}]$
3     **for** $temp$ *in* $Sam_{up}$ **do**
4        **if** $temp \in [pf_{idx} - \vartheta, pf_{idx} + \vartheta]$ **then**
5           **add** $temp$ in array $pf^{new}_{idx}$
6        **end**
7     **end**
8     $pf^{new}_s \leftarrow pf^{new}_{idx}[0], pf^{new}_e \leftarrow pf^{new}_{idx}[-1]$
9     **for** $i$ *in* $[pf^{new}_s, pf^{new}_e]$ **do**
10       $t \leftarrow i, p^c_t \leftarrow p^c_{t_{ori}}$
11       **add** $(t, p^c_t)$ in $P^c$
12     **end**
13 **end**
14 **return** $P^c$

---

## B    ALGORITHM FOR MINING PSEUDO LABELS

In WS-TAL, pseudo label based methods (Luo et al., 2020; Yang et al., 2021; Pardo et al., 2021; Zhai et al., 2020) are widely used to obtain segmented pseudo labels to bridge the gap between classification and localization. Previous detection results are usually extended to generate pseudo labels. Due to the limited information in each segment, pseudo labels can obtain over-complete or incomplete proposals. In Sec. 3.3 of this paper, Our proposed mining algorithm can utilize the precise location information provided by point-level labels, which can generate more accurate pseudo labels.

Algorithm 2 explains the mining process. Firstly, between two adjacent action instances, we set a threshold of $\gamma_{bkg}$=0.85 to filter out noise segments and select segments with background scores $b_t$ greater than this threshold to generate pseudo background point-level labels. If no such segments exist, we set the background labels at the index with the highest $b_t$. After generating these pseudo background point-level labels, we search a few possible pseudo action frames from each adjacent action frame and pseudo background frame. Specifically, the scores $b_t$ of pseudo action frames must be lower than the threshold $\gamma_{act}$=0.25, and its action score is the highest value in $\widehat{Q}$.

Figure 4 visualizes two groups of pseudo label sequences from our method and LACP (Lee & Byun, 2021), respectively. Obviously, our mined pseudo labels cover more complete instances. In particular, for the generation of pseudo action labels, our method is significantly more effective than LACP.

---

**Algorithm 2:** Mining Pseudo Point-level labels

---

**input** : the original action point-level labels: $P^{act} = \{(t_i, p_{t_i})\}_{i=1}^{N^{act}}$, the background score $B \in \mathbb{R}^T$, the threshold of the potential background: $\gamma_{bkg}$, the threshold of the potential action: $\gamma_{act}$, the final score $\widehat{Q}$

**output**: pseudo background point-level labels: $P^{bkg} = \{t_j\}_{j=1}^{N^{bkg}}$, pseudo action point-level labels: $P_{pse}^{act} = \{(t_k, p_{t_k})\}_{k=1}^{N_{pse}^{act}}$

1 **Initialize** create a new array with the same dimension size as $B$ and all values of 0: $P^{bkg}$
2 **Set** index of the background from $P^{bkg}$ as $I^{bkg}$ and index of the action from $P^{act}$ as $I^{act}$
   *// Mark the indexes with the background score $b_t$ higher than $\gamma_{bkg}$ as the pseudo background labels*
3 **for** $i = 1$ *to* $T$ **do**
4     **if** $b_i > \gamma_{bkg}$ **then**
5        **add** $i$ to $I^{bkg}$, and $P^{bkg}[i] = 1$
6     **end**
7 **end**
   *// If there is no background label between adjacent action points, select the index with the highest background score $b_t$ within the interval as the pseudo background label*
8 $left \leftarrow [-1, I^{act}[:-2]], right \leftarrow [I^{act}[1:], T]$
9 **for** $idx_{left}, idx_{right}$ *in* $zip(left, right)$ **do**
10     **for** $j$ *in* $I^{bkg}$ **do**
11        **if** $idx_{left} < j < idx_{right}$ **then**
12           **go to** step 9
13        **end**
14     **end**
15     find the index $idx_{temp}$ of maximum score in $B[idx_{left} : idx_{right}]$, and $P^{bkg}[idx_{temp}] = 1$
16     **add** $idx_{temp}$ to $I^{bkg}$
17 **end**
   *// Generate continuous background segments*
18 **for** $idx_{left}^{bkg}, idx_{right}^{bkg}$ *in* $zip(I^{bkg}[:-2], I^{bkg}[1:])$ **do**
19     **for** $j$ *in* $I^{act}$ **do**
20        **if** $idx_{left}^{bkg} < j < idx_{right}^{bkg}$ **then**
21           **go to** step 18
22        **end**
23     **end**
24     $P^{bkg}[idx_{left}^{bkg} : idx_{right}^{bkg}] = 1$, and **add** $[idx_{left}^{bkg} : idx_{right}^{bkg}]$ to $I^{bkg}$
25 **end**
   *// Search for pseudo action point-level labels*
26 **for** $i = 1$ *to* $T$ **do**
27     **if** $b_i \leq \gamma_{act}$ *and* $i \notin I^{bkg}$ **then**
28        nearest left action $(t_{left}, p_{t_{left}})$ and right action $(t_{right}, p_{t_{right}})$ for index $i$, where $p_{t_{left}}, p_{t_{right}}$ are action labels and $t_{left} < i < t_{right}$
29        **if** *not exist* $j \in I^{bkg}$ *is satisfied* $t_{left} < j < i$ **then**
30           **if** the score of category $c$, $q_i^c = \widehat{Q}[i : c]$ is highest for $\widehat{Q}[i :]$ **then**
31              **add** $(i, p_{t_{left}})$ in $P^{act}$, and $N^{act} \leftarrow N^{act} + 1$
32        **end**
33        **if** *not exist* $j$ *in* $I^{bkg}$ *is satisfied* $i < j < t_{right}$ **then**
34           **if** the score of category $c$, $q_i^c = \widehat{Q}[i : c]$ is highest for $\widehat{Q}[i :]$ **then**
35              **add** $(i, p_{t_{right}})$ in $P^{act}$, and $N^{act} \leftarrow N^{act} + 1$
36        **end**
37     **end**
38 **end**
39 $P_{pse}^{act} \leftarrow P^{act}$
40 **Return** $P^{bkg}, P_{pse}^{act}$

---

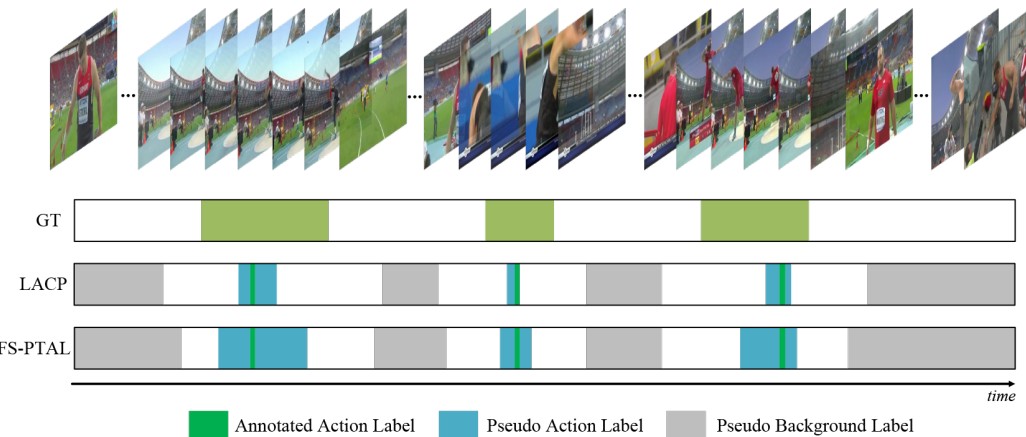

Figure 4: Label visualization comparison with LACP on THUMOS14. We provide an example from the training data *"video_validation_0000784"*. The action instance in this example is *Shotput*. The pseudo labels generated from LACP and our FS-PTAL as well as ground-truth are visualized in this figure.

Table 3: The influence of different settings on several groups of hyper-parameters on THUMOS14. We only present a portion of the experimental results for hyper-parameter selection in this table. mAP@AVG is the averaged mAP at the thresholds 0.1:0.1:0.7. That $\tau_*$ is [2.5, 1.67] indicates that $\tau_1$ and $\tau_2$ are respectively 2.5 and 1.67.

| $\tau_*$ | [2.5, 1.67] | [2.5, 2] | [2.5, 1.5] | [2, 1.67] | [3, 1.67] |
|---|---|---|---|---|---|
| mAP@AVG(%) | **56.73** | 56.49 | 56.62 | 56.22 | 56.27 |
| $\psi_{same}$ | 0.75 | 0.8 | 0.85 | 0.9 | 0.95 |
| mAP@AVG(%) | 55.67 | 56.02 | 56.38 | 56.09 | **56.73** |
| $\psi_{diff}$ | 0.05 | 0.1 | 0.15 | 0.2 | 0.25 |
| mAP@AVG(%) | 56.64 | **56.73** | 56.27 | 55.81 | 55.52 |

Table 4: State-of-the-art comparison on ActivityNet 1.3 (Heilbron et al., 2015). AVG is the averaged mAP at the thresholds 0.5:0.05:0.95.

| Supervision | Method | mAP@IoU(%) | | | AVG |
|---|---|---|---|---|---|
| | | 0.5 | 0.75 | 0.95 | |
| Video-level | Bas-Net (Lee et al., 2020) | 34.5 | 22.5 | 4.9 | 22.2 |
| | TS-PCA (Yang et al., 2021) | 37.4 | 23.5 | 5.9 | 23.7 |
| | FAC-Net (Huang et al., 2021) | 37.6 | 24.2 | 6.0 | 24.0 |
| | ACM-Net (Qu et al., 2021) | 40.1 | 24.2 | 6.2 | 24.6 |
| | ASM-Loc (He et al., 2022) | 41.0 | 24.9 | 6.2 | 25.1 |
| | FTCL (Gao et al., 2022) | 40.0 | 24.2 | 6.4 | 24.8 |
| | CASE (Liu et al., 2023) | **43.2** | 26.2 | **6.7** | 26.8 |
| Point-level | LACP (Lee & Byun, 2021) | 40.4 | 24.6 | 5.7 | 25.1 |
| | Ours | 43.1 | **26.9** | 6.1 | **27.3** |

Table 5: State-of-the-art comparison on BEOID (Damen et al., 2016) and GTEA (Lei & Todorovic, 2018). AVG denotes the average mAP at the thresholds 0.1:0.1:0.7.

| Dataset | Method | mAP@IoU(%) | | | | AVG |
|---|---|---|---|---|---|---|
| | | 0.1 | 0.3 | 0.5 | 0.7 | |
| GTEA | SF-Net (Ma et al., 2020) | 58.0 | 37.9 | 19.3 | 11.9 | 31.0 |
| | Ju *et al.* (Ju et al., 2020) | 59.7 | 38.3 | 21.9 | 18.1 | 33.7 |
| | LACP (Lee & Byun, 2021) | 63.9 | 55.7 | 33.9 | 20.8 | 43.5 |
| | Ours | **65.3** | **56.8** | **34.4** | **21.2** | **44.3** |
| BEOID | SF-Net (Ma et al., 2020) | 62.9 | 40.6 | 16.7 | 3.5 | 30.9 |
| | Ju *et al.* (Ju et al., 2020) | 63.2 | 46.8 | 20.9 | 5.8 | 34.9 |
| | BackTAL (Yang et al., 2022) | 60.1 | 40.9 | 21.2 | 11.0 | 32.5 |
| | LACP (Lee & Byun, 2021) | 76.9 | 61.4 | 42.7 | 25.1 | 51.8 |
| | Ours | **78.1** | **62.2** | **43.6** | **25.4** | **52.5** |

Table 6: State-of-the-art comparison using different point-level labels on THUMOS14. Where † indicates the use of labels manually annotated in (Ma et al., 2020), ‡ denotes the use of automatically marked labels from (Moltisanti et al., 2019).

| Method | mAP@IoU(%) | | | | | | | AVG (0.1:0.5) | AVG (0.3:0.7) |
|---|---|---|---|---|---|---|---|---|---|
| | 0.1 | 0.2 | 0.3 | 0.4 | 0.5 | 0.6 | 0.7 | | |
| SF-Net† (Ma et al., 2020) | 71.0 | 63.4 | 53.2 | 40.7 | 29.3 | 18.4 | 9.6 | 51.5 | 30.2 |
| Ju *et al.*† (Ju et al., 2020) | 72.8 | 64.9 | 58.1 | 46.4 | 34.5 | 21.8 | 11.9 | 55.3 | 34.5 |
| LACP† (Lee & Byun, 2021) | 75.1 | 70.5 | 63.3 | 55.2 | 43.9 | 33.3 | 20.8 | 61.6 | 43.3 |
| Ours† | **79.3** | **73.8** | **67.5** | **60.2** | **49.4** | **37.6** | **23.1** | **66.0** | **47.6** |
| SF-Net‡ (Ma et al., 2020) | 68.3 | 62.3 | 52.8 | 42.2 | 30.5 | 20.6 | 12.0 | 51.2 | 31.6 |
| Ju *et al.*‡ (Ju et al., 2020) | 72.3 | 64.7 | 58.2 | 47.1 | 35.9 | 23.0 | 12.8 | 55.6 | 35.4 |
| LACP‡ (Lee & Byun, 2021) | 75.7 | 71.4 | 64.6 | 56.5 | 45.3 | 34.5 | 21.8 | 62.7 | 44.5 |
| Ours‡ | **79.1** | **74.6** | **68.6** | **61.0** | **50.7** | **38.9** | **24.2** | **66.8** | **48.7** |

## C    RELATED SETTING AND INFERENCE

We employ I3D (Zeng et al., 2019) networks pre-trained on Kinetics-400(Zeng et al., 2019) to extract feature and apply the TV-L1 algorithm (Huang et al., 2022) to extract optical flow from RGB frames. For THUMOS14, ActivityNet 1.3, GTEA and BEOID, our model is optimized by Adam (Luo et al., 2020) with a learning rate of 0.0001 and batch sizes of 16, 64, 8 and 8, respectively. Meanwhile, the training epoch on these four benchmarks are 1500, 60, 200 and 200.

The hyper-parameters in this paper are determined by grid search, the specific settings are as follows. The sampling weights $\tau_1$ and $\tau_2$ are 2.5 and 1.67, respectively. The threshold $\psi_{same}$ and $\psi_{diff}$ are 0.9 and 0.1. In Table 3, we show the influence of different settings of the above groups of hyper-parameters on the results. Numerous comparative experiments can demonstrate that our hyper-parameter settings are reasonable. Specifically, the weighted parameter $\lambda_1$, $\lambda_2$, $\lambda_3$ and $\lambda_4$ are all 1, these are determined based on massive previous works and experience.

Table 7: The comparison of different calculation methods for OIC Loss on THUMOS14. $\triangle$ denotes the use of the initial calculation(Shou et al., 2018) about OIC Loss. $\#$ denotes the user of our optimized calculation for OIC Loss.

| Method | mAP@IoU(%) | | | | AVG (0.1:0.7) |
|---|---|---|---|---|---|
| | 0.1 | 0.3 | 0.5 | 0.7 | |
| FS-PTAL$^{\triangle}$ | 78.4 | 67.5 | 49.2 | 23.2 | 55.8 |
| FS-PTAL$^{\#}$ | **79.1** | **68.6** | **50.7** | **24.2** | **56.7** |

In the inference process, we first determine which action categories are to be localized by thresholding the video-level classification score $Q_v$ with $\theta_{vid}$=0.25. Afterwards, we threshold at the final score $\widehat{Q}$ to select candidate segments. And after a large number of consecutive candidates merging into many action proposals, we use non-maximum suppression (NMS=0.7) to remove overlapping proposals.

## D    MORE EXPERIMENTAL RESULTS AND ANALYSIS

As shown in Table 4, our FS-PTAL obtain the absolute average mAP gains of 0.5% on ActivityNet 1.3 compared to the Liu et al. (2023). Notably, at mAP@0.95, we have no improvement, even weaker than a few video-level methods(weakly-supervised). We believe that ActivityNet 1.3 dataset contains many action instances whose lengths are close to the duration of the input video. However, our mined pseudo background labels greatly affect the complete localization of such ultra long action instances, resulting the performance does not improve under IoU=0.95 conditions. From Table 5, we can see that our method has achieved a new state-of-the-art on GTEA and BEOID, acquiring the absolute mAP gains of 0.8% and 0.7% compared to the previous advanced methods, respectively.

In Table 6, we present the experimental results using two different point-level labels (Moltisanti et al., 2019; Ma et al., 2020). The results show that regardless of the type of labels, our FS-PTAL is significantly superior to the previous methods, indicating the robustness of our method. In addition, we speculate that sampling distribution update method in (Moltisanti et al., 2019) make automatically marked labels more likely to focus on areas with significant changes in action instances compared to manually annotated labels (Ma et al., 2020), so there is a slight improvement in performance using automatically marked labels.

In table 7, we compare the effects of original OIC calculation and optimized OIC calculations on experimental results. Obviously, our optimized OIC calculation method has a 0.9% performance improvement on THUMOS14. As in Sec. 3.4, the initial calculation does have drawbacks in videos of alternating long and short actions.

## E    DETAILED PRESENTATION FOR QUALITATIVE COMPARISON AND VISUALIZATION

In Figures 5 and 6, we visualize the comparison of detection results between our FS-PTAL and DELU (Chen et al., 2022) and LACP (Lee & Byun, 2021) on THUMOS14, respectively. As shown, the detection results produced by DELU are often fragmented, because DELU only uses video-level label, the obtained location information is extremely limited. In contrast, the detection results of LACP are relatively complete, but still accompanied by interference from noise segments. Compared with the above two methods, our FS-PTAL to some extent eliminates noise segments, making the detection results more complete and more in line with GT. In Figure 7, we conduct the performance analyses of BackTAL (Yang et al., 2022), ASM (He et al., 2022), LACP and our FS-PTAL. It is simple to know that our model has the highest TP(true positive) rate, and this means that the accuracy of our model's locating actions does indeed significantly improve compared to previous methods. Meanwhile, the error rate of our method also decrease obviously, indicating that the problem about the action-background confusion is somewhat mitigated and our FS-PTAL does indeed better provide guidance on fine-grained separation of action-background, significantly improving performance.

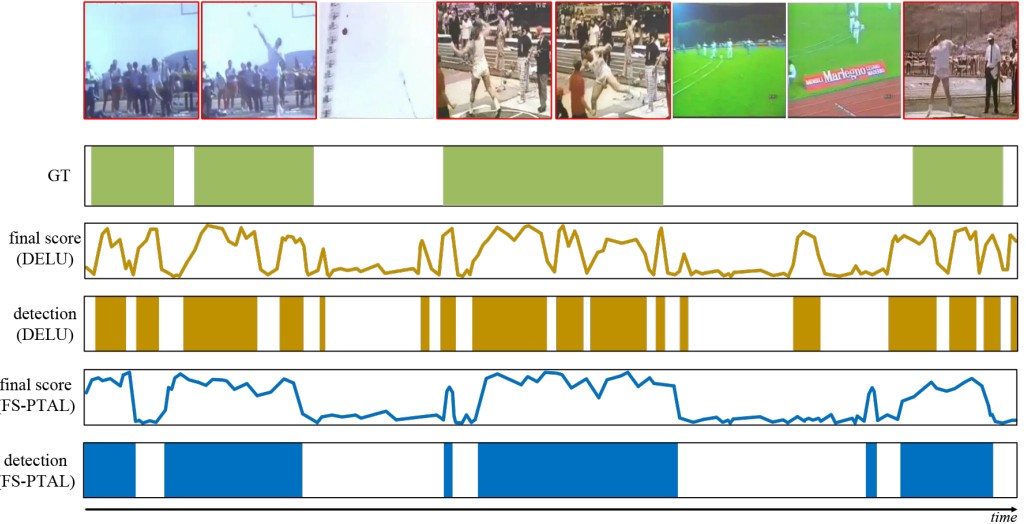

Figure 5: Qualitative comparison with DELU (Chen et al., 2022) on THUMOS14. We provide an example from the test data *"video_test_0000129"*. The action instance in this example is *Shotput*. The final scores and detection results from DELU and our FS-PTAL as well as ground-truth are visualized in this figure. The red boxes indicate that the frames are from action instances.

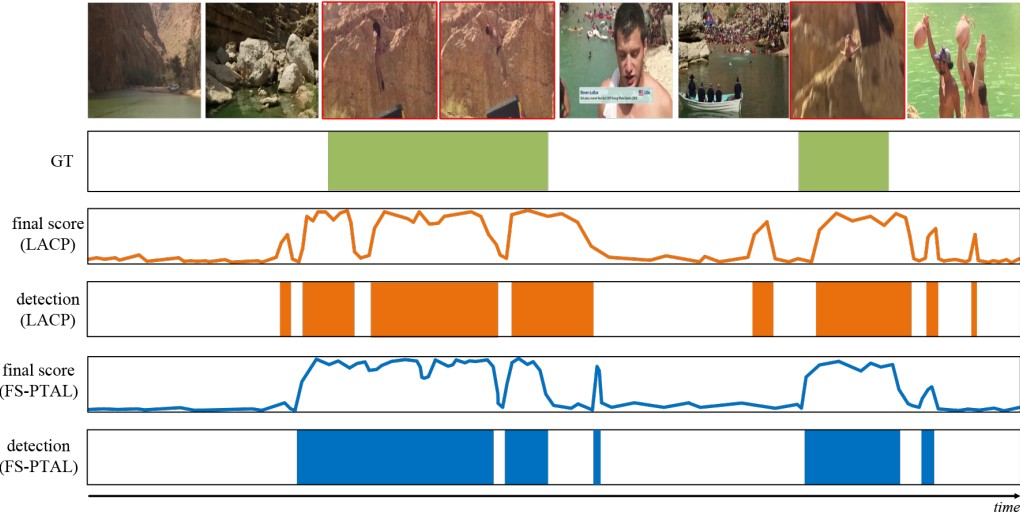

Figure 6: Qualitative comparison with LACP (Lee & Byun, 2021) on THUMOS14. We provide an example from the test data *"video_test_0000173"*. The action instance in this example is *CliffDiving*. The final scores and detection results from LACP and our FS-PTAL as well as ground-truth are visualized in this figure. The red boxes indicate the frames are from action instances.

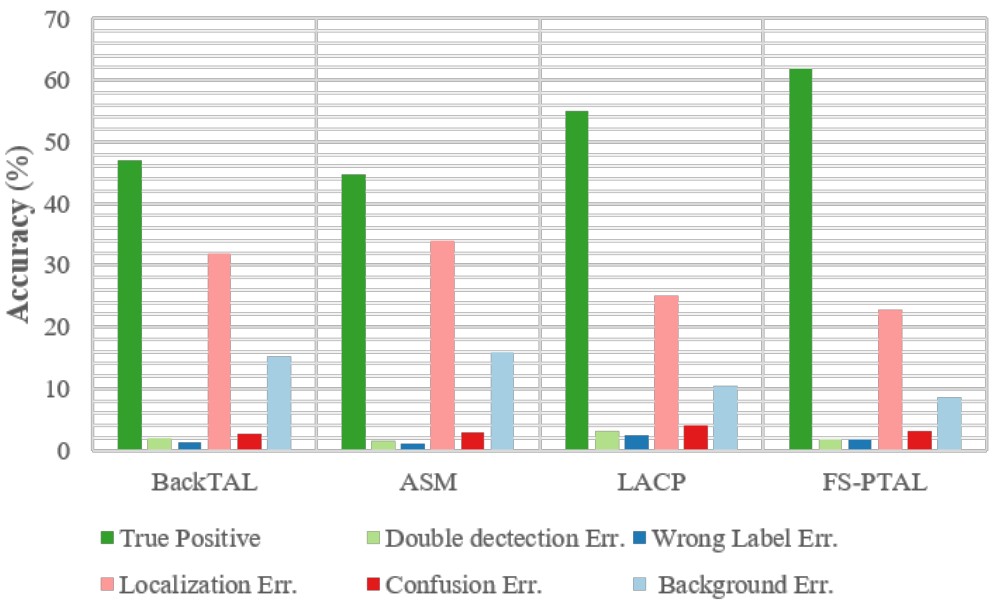

Figure 7: The performance comparison of Top 1 Prediction at IoU=0.5 on THUMOS14. We utilize diagnostic tools (Alwassel et al., 2018) to analyze our FS-PTAL and other three state-of-the-art methods, and visualize the diagnosing results.

