# OpenReview forum: "Fine-grained Separation of Action-Background for Point-Level Temporal Action Localization"
_ICLR.cc/2024/Conference — Submitted to ICLR 2024_

### Official Review · Reviewer_L6ip · 2023-10-30

**Soundness:** 3 good
**Presentation:** 3 good
**Contribution:** 3 good
**Rating:** 5
**Confidence:** 5

**Summary:**

This paper proposes a framework to mine pseudo point-level labels for improving the performance of weakly supervised temporal action localization. There are four components, including label extension, pseudo label mining, score contrast module and feature separation module. Extensive experiments on four benchmarks verify the state-of-the-art performance of the proposed framework.

**Strengths:**

The proposed method is well-motivated with the error analysis and is also technically sound. Extensive experiments on four benchmarks verify the state-of-the-art performance of the proposed framework.

**Weaknesses:**

1. The major drawback of this paper is its incremental novelty. The proposed components are all modified versions of the off-the-shelf methods, for example, label extension originates from dynamic sampling, pseudo label mining modifies the one of LACP [1], score contrast module and feature separation module also borrow the idea of OIC loss [2] and Co-Activity Similarity [3].
2. Some important references are missing, for example, Zhou et al. [4] also explore generating high-quality pseudo labels for weakly supervised temporal action localization.
3. More ablation studies are needed, for example, the performance with a single proposed component.
4. Qualitative results are needed to show the performance.

[1] Learning action completeness from points for weakly-supervised temporal action localization, ICCV 2021.

[2] Autoloc: Weakly supervised temporal action localization in untrimmed videos, ECCV 2018.

[3] W-talc: Weakly-supervised temporal activity localization and classification, ECCV 2018.

[4] Improving Weakly Supervised Temporal Action Localization by Bridging Train-Test Gap in Pseudo Labels, CVPR 2023.

**Questions:**

1. There are too many hyper-parameters in the proposed method, which may increase the difficulty of reproduction. How much would these hyper-parameters affect the model? Please show more sensitivity analysis of hyper-parameters.
2.  In Formula 1, why $τ_2$ is only assigned to the nearest frame of the boundary but not the outer segment like OIC loss?

---

> ### Author Response · Authors · 2023-11-22
>
> Thank you for your question. I will answer them one by one.
>
> **Q1: The major drawback of this paper is its incremental novelty.**
>
> **A1**: I will emphasize the technical novelty of this paper. The overall two-step framework does indeed refer to LACP to some extent, but there are significant differences. Firstly, we take the lead in designing label extension module and feature up-sampling method to enhance action features in point-level supervision, using location information provided by point labels. Secondly, our proposed feature separation module is also novel and it is worth noting that this is different from the Co-Activity Similiarity of W-TALC. The role of Co-Activity Similiarity is to determine the correlation between similar categories of videos, in contrast, our feature separation module extracts the corresponding action feature matrix and background feature matrix, and calculates the feature separation loss by using the cosine similarity, to amplify the differences between action-backgrounds and enhance the similarity between actions or backgrounds. This calculation method for feature separation has not been proposed before, and our experimental results show that it is successful. Besides, we optimized the pseudo label mining algorithm in LACP, enhancing performance of mining pseudo action labels. Finally, we make a correction to the overlapping issue between action instances during loss calculation for practical scenarios, although this is only a minor change, it is a problem discovered through extensive experimental verification, and as shown in Table 7, the performance improvement it brings is also significant. We believe that this correction has good implications for many methods in this field.
>
> **Q2: Some important references are missing and more ablation studies are needed.**
>
> **A2**: We have added some of the latest articles you mentioned to the related work. And we demonstrated the effectiveness of our proposed modules or methods through ablation experiments in Table 2. Firstly, in Sec. A of appendix, we provide a detailed description of the algorithm for the label extension module, which relies on the rough action positions provided by point labels for feature upsampling and label extension. Intuitively, this will definitely lead to performance improvement. In future paper versions, we can align and visualize labels with action features to prove the accuracy of label extension in the temporal dimension. Besides, in Figure 3, we visualize the range selection of the pre-and post-optimizatized outer-inner contrast score, and it is obvious that our optimized calculation method is more accurate and can magnify the difference between the action-background segments for practical scenarios. Meanwhile, Figure 4 visualizes the comparison between our FS-PTAL and LACP in pseudo label mining methods, it is evident that the pseudo labels mined by our method are more closely aligned with GT and have a wider coverage area, indicating that our optimized method for label mining is more effective. Finally, the effectiveness of the feature separation module is also intuitive, by mapping to the feature embedding space, all action feature matrices and background feature matrices are extracted to calculate the cosine similarity and the feature separation loss, achieving action-background separation. Table 2 shows that the impact of the feature separation module on the experimental results is significant.
>
> **Q3: There are too many hyper-parameters in the proposed method, which may increase the difficulty of reproduction.**
>
> **A3**: In Sec. C of the appendix, we provide detailed training hyper-parameter settings. Numerous comparative experiments can demonstrate that our hyper-parameter settings are reasonable. In Table 3, we show the influence of different settings of several hyper-parameters on the results. If necessary, we can add more hyper-parameter comparison results in subsequent versions. Specifically, the weighted parameter $λ_1$, $λ_2$, $λ_3$ and $λ_4$  are all 1, these are determined based on massive previous works and experience.
>
> **Q4: In Formula (1), why $τ_2$ is only assigned to the nearest frame of the boundary but not the outer segment like OIC loss?**
>
> **A4**: Formula (1) in the label extension module, which belongs to the data preprocessing stage. We determine the rough position of the action based on the point-level labels and assign fixed sampling weights $τ_1$ and $τ_2$ within and around the actions. The outer segment like OIC loss need to undergo convolution operations and class activation functions, and filter out action segments. Obviously, this operation undoubtedly wastes a lot of time and is very complex. Our label extension module was originally in the preprocessing stage, so directly using the rough information of point-level labels is quite intuitive.

---

> > ### Comment · Reviewer_L6ip · 2023-11-23
> > **Post-rebuttal comments**
> >
> > I appreciate the authors' response.
> >
> > The authors emphasize their novelty in the rebuttal, which is however not convincing to me. The authors acknowledge that their framework draws inspiration from LACP, essentially representing an enhanced version of it. Moreover, while each component they propose is linked to existing works and has been modified by them, the reasoning behind their design choices remains somewhat unclear.
> >
> > For Table 2, the authors have not yet included an ablation study for each individual component. In addition, for Q4, the response cannot convince me without a quantitative comparison.
> >
> > Due to these factors, I have decided to maintain my rating as it is.

---

### Official Review · Reviewer_KUWp · 2023-10-31

**Soundness:** 2 fair
**Presentation:** 1 poor
**Contribution:** 2 fair
**Rating:** 3
**Confidence:** 5

**Summary:**

The paper tackles the problem of weakly supervised temporal action detection with point-level supervision. The proposed method follows a similar pipeline to LACP and introduces design on label extension, pseudo label mining, feature separation and score contrasting to improve detection performance. Extensive experiments on popular benchmarks show improvements based on previous state-of-the-art point-level weakly-supervised methods.

**Strengths:**

The proposed method achieves non-trivial improvement on Thumos14 compared to previous weakly supervised method. The ablation study supports the effectiveness of each component in the framework.

**Weaknesses:**

- The proposed method looks incremental on LACP (Lee & Byun (2021)). The overall pipeline, pseudo label mining, feature separation and score contrast module are very similar to LACP.  The technical contributions look like modifications to each LACP component from the engineering side. There should be more in-depth analysis to explain the motivation of each proposed design.
- Experiments:
  - Missing important benchmark results (ActivityNet) in the main paper.
  - Missing important methods in comparison table (tab.4). Is there a reason to not compare with other point-level supervision methods (SF-Net, BackTAL and Ju et.al. ) method on ActivityNet?
  - The performance on Activitynet against video-level supervision methods does not look competitve. If this is due to the sparsity of action in activitynet videos, then the authors should provide  comparison with video-level approaches on other benchmarks (GTEA and BEOID), as GTEA and BEOID has denser action distributions and should be better according to the authors' claim.
- Generally, the paper suffers from bad writing, including grammatical errors (subject-verb agreement, verb tense, etc), confusing wording, poor organization (introduction too long, and experiments too short), and inconsistent citation style (e.g. for pointTAD). The overall paper feels repetitive and lacks flow. The authors need to revise the whole paper thoroughly and properly organize the content in the main paper (eg. move important comparison results from supp. to main paper).
- There's a factual error in Introduction, Related Work and Comparison table, that pointTAD is in fact not a weakly supervised approach to TAD, but a fully-supervised method.

**Questions:**

- Figure 3 is confusing and lacking important captions. The meaning of R(S_c) and how to compute the inner and outer score is not clear in the figure. Although section 3.4 seems to explain figure 3, we don't see consistent notations in the text and figure, for example the notations in text (s_n^c , e_n^c  and R(SL_c)) do not have corresponding illustration in the figure.

---

> ### Author Response · Authors · 2023-11-22
>
> Thank you for your question. I will answer them one by one.
>
> **Q1: The proposed method looks incremental on LACP.**
>
> **A1**: I will emphasize the technical novelty of this paper. The overall two-step framework does indeed refer to LACP to some extent, but there are significant differences. Firstly, we take the lead in designing label extension module and feature up-sampling method to enhance action features in point-level supervision, using location information provided by point labels. Secondly, our proposed feature separation module is also novel, we extract and merge the positions marked by point labels to generate the action point feature matrix and the background point feature matrix. We use cosine similarity to amplify the differences between action-backgrounds and enhance the similarity between actions or backgrounds. This calculation method for feature separation has not been proposed before, and our experimental results show that it is successful. Besides, we optimized the pseudo label mining algorithm in LACP, enhancing performance of mining pseudo action labels. Finally, we make a correction to the overlapping issue between action instances during loss calculation for practical scenarios, although this is only a minor change, it is a problem discovered through extensive experimental verification, and as shown in Table 7, the performance improvement it brings is also significant. We believe that this correction has good implications for many methods in this field.
>
> **Q2: Regarding the experiments.**
>
> **A2**: We have experimental results on the ActivityNet1.3 in Table 4, but we did not compare with other point-level supervision methods (SF-Net, BackTAL and Ju et.al. ) because hese methods either have poor performance on ActiviNet1.3 (for example, the mean Map@[0.5,0.95] of SF-Net is only 21.2) or lack parameters trained on this dataset. All in all, we only select the most representative point supervised method LACP for comparison. Besides, we would like to provide experimental results of video-level methods on GTEA and BEOID, but all video-level methods do not provide training parameters on these two benchmarks. We attempted to debug the source code of these video-level methods on these two benchmarks, but did not achieve success. Therefore, we only present the results of point-level methods on these two benchmarks.
>
> **Q3: The paper suffers from bad writing, including grammatical errors (subject-verb agreement, verb tense, etc), confusing wording, poor organization (introduction too long, and experiments too short), and inconsistent citation style (e.g. for pointTAD).**
>
> **A3**: We provide explanations for writing and PointTAD. If this paper can pass, in future versions of the paper, we will reduce the description of introduction and related work, and move the presentation of Figure 1 to the appendix. We will add more experimental tables in the main text section. Finally, regarding the description of PointTAD, we do have a minor error, and we have removed its description from Table 1 and related work.
>
> **Q4: Figure 3 is confusing and lacking important captions.**
> **A4**: We have modified the incorrect $R (S_c)$ to $R (SL_c)$, so there will be corresponding illustration in the Figure 3. Regarding the specific calculation process of $R (SL_c)$, the steps are as follows: Firstly, we calculate the segment length $l_{seg}$ from $s_n^c$ and $e_n^c$. Next, we select the external calculation range according to formula (8). Finally, the calculation of $R (SL_c)$ is based on the following formula. For the convenience, we define the length on the left and right side of the external range as $l_{left}$, $l_{right}$, respectively.
>
> $R (SL_c)=\frac{1}{N_{s l}^{c}}\sum_{n=1}^{N_{s l}^{c}}(\frac{1}{l_{\text {seg}}}\sum_{t=s_{n}^{c}}^{e_{n}^{c}} \widehat{q_{t}}[c]-\frac{1}{l_{\text{left}}+l_{\text {right}}}(\sum_{t=\min(s_{n}^{c}-\delta l_{seg}, e_{n-1}^{c})}^{s_{n}^{c}-1} \widehat{q}_{t}[c]$
>
> $+\sum_{t=e_{n}^{c}+1}^{\min(e_{n}^{c}+\delta l_{seg}, s_{n+1}^{c})} \widehat{q}_{t}[c]))$

---

> > ### Comment · Reviewer_KUWp · 2023-11-23
> > **Post-rebuttal Comment**
> >
> > I appreciate the authors efforts in providing the response, but my concerns with the novelty remain unrelieved. Each component in the framework has its counterpart in previous works and the improvement merely looks like technical modifications. The insights in proposing these modules remain similar to previous works. Furthermore, I agree with reviewer PXaQ and L6ip on the need for an apple-to-apple ablation study. Based on the above reasons, the paper in its current form does not meet the standards of ICLR, therefore I keep my original rating.

---

### Official Review · Reviewer_8Huu · 2023-10-31

**Soundness:** 3 good
**Presentation:** 3 good
**Contribution:** 3 good
**Rating:** 6
**Confidence:** 4

**Summary:**

In this paper, the author proposes a new method for point-level temporal action localization. The proposed method utilizes multiple modules including label extension module, pseudo label mining and score contrast module to enhance the performance of point-level supervised temporal action localization. The proposed method achieves performance gain over the standard benchmark temporal action localization datasets.

**Strengths:**

In general, I think this paper has clear definitions, good illustrations, and exhaustive experiments to verify the effectiveness of the proposed method. The proposed method has superior performance on major benchmark datasets.

**Weaknesses:**

However, I still have some little concerns about this paper:

1. The writing should be polished. There are some grammatical errors like "genearting" on page 5. Also, some abbreviations should be re-introduced like OIC on page 6, though the author introduces it on page 2. Also, all formulas should end with a comma or period, and space after the bracket, etc.

2. I think the experiment part could be revised to provide a clear comparison. First, in Table 1, the author could provide provides more recent fully supervised temporal action localization methods like ActionFormer, TriDet, etc. They can easily achieve around 66+ mean mAP@[0.3,0.7] on THUMOS14. Don't claim those state-of-the-art methods will make the result table not convincing. Also, Table 1 reports mean mAP@[0.1,0.5] and mean mAP@[0.3,0.7]. While in Table 2, the author only reports the mean mAP@[0.1,0.7]. It is confusing here. Also, Table 2 should become a step-to-step ablation, the current form is somehow weird.

**Questions:**

Please mainly see the weaknesses section for details.

---

> ### Author Response · Authors · 2023-11-22
>
> Thank you for your question. I will answer them one by one.
>
> **Q1: The writing should be polished.**
>
> **A1**: We have corrected the two errors you raised. If this paper can pass, we will polish the writing and pay attention to the formatting of all formulas in subsequent versions of the paper.
>
> **Q2: The experiment part could be revised to provide a clear comparison.**
>
> **A2**: We think your proposed modification of the experimental section is clear. There are no recent temporal action localization methods such as ActionFormer and TriDet in Table 1, because these methods are based on the transformer structure, while the methods which we compare are mainly based on convolutional architecture. These methods have smaller models, fewer parameters, and lower costs. In Table 2, we believe that the mean mAP@[0.1,0.7] is sufficient to identify the performance gap. The reason why Table 1 is divided into the mean mAP@[0.1,0.5] and the mean mAP@[0.3,0.7] is because some methods only have the former and some methods only have the latter, which is not enough to calculate the mean mAP@[0.1,0.7]. Therefore, we use two standards to display performance. Meanwhile, for the ablation experiment in Table 2, we refer to previous work. We delete the proposed innovative modules or methods separately to observe the effect of each module or method on performance.

---

> > ### Comment · Reviewer_8Huu · 2023-11-23
> > **Postrebuttal comments**
> >
> > After reading the author's rebuttal and reviews from other reviewers, I would like to keep my original rating. The author's rebuttal doesn't convince me about the experiment part, also I raise some new concerns about the novelty part/hyperparameter part.

---

### Official Review · Reviewer_PXaQ · 2023-11-01

**Soundness:** 2 fair
**Presentation:** 3 good
**Contribution:** 1 poor
**Rating:** 3
**Confidence:** 5

**Summary:**

This paper address the task of temporal action localization under point-level supervision. The authors focus on the observation that existing approaches have difficulty in discriminating the action and background, leading to significant localization and background errors. To tackle this, the authors introduce FS-PTAL, a new framework that aggregates pseudo labels based on sparse point-level annotations and enhances the contrast between the action and background. Experiments on benchmark datasets confirm the superiority of the proposed model over existing state-of-the-arts.

**Strengths:**

+ The manuscript is overall well-organized and easy to follow.
+ The motivation behind the work is clear and reasonable; enlarging the discrepancy between action and background frames is the key challenge in the weakly-supervised setting.
+ The proposed model surpasses the prior arts by non-trivial margins, which manifests its effectiveness well.

**Weaknesses:**

- The technical novelty of the paper is limited. The overall two-step framework strictly follows that of LACP (Lee & Byun, 2021), with improvements made to the original loss (i.e., Feature Separation loss), a correction to the overlapping issue between action instances during loss calculation (i.e., Score Contrast loss), and addition of new elements (i.e., Label Extension Module). While these contribute to the paper, from my view, this work seems an extension of the previous work (Lee & Byun, 2021), and the newly introduced contributions are slightly under the standard bar of top-tier conferences.
- This paper lacks comprehensive analyses to substantiate the effectiveness of the proposed components. As noted in the above weakness, the model improves the previous approach with modifications and additions. However, their actual effects and how they help are not analyzed in the experiments. Also, apple-to-apple comparisons with the original method would be desirable.
- The paper is not self-contained in its current form, distracting the readers by making them alternate between the main paper and the appendix. Also, only the two kinds of experimental results are provided in the manuscript, while the remaining ones are in the appendix. It is strongly encouraged for the authors to trim the inappropriately long content in Introduction and Related work (e.g., Figure 1 occupies too much space), and add more experimental results in the main text.

(Minor)

The reference format to PointTAD is wrong; it should be formatted as (Tan et al., 2022). Additionally, PointTAD is not a weakly-supervised approach, so the comparison with it in Table 1 is inappropriate.

**Questions:**

Please refer to Weakness section.

---

> ### Author Response · Authors · 2023-11-22
>
> Thank you for your question. I will answer them one by one.
>
> **Q1: The technical novelty of the paper is limited.**
>
> **A1**: I will emphasize the technical novelty of this paper. The overall two-step framework does indeed refer to LACP to some extent, but there are significant differences. Firstly, we take the lead in designing label extension module and feature up-sampling method to enhance action features in point-level supervision, using location information provided by point labels. Secondly, our proposed feature separation module is also novel, we extract and merge the positions marked by point labels to generate the action point feature matrix and the background point feature matrix. We use cosine similarity to amplify the differences between action-backgrounds and enhance the similarity between actions or backgrounds. This calculation method for feature separation has not been proposed before, and our experimental results show that it is successful. Besides, we optimized the pseudo label mining algorithm in LACP, enhancing performance of mining pseudo action labels. Finally, we make a correction to the overlapping issue between action instances during loss calculation for practical scenarios, although this is only a minor change, it is a problem discovered through extensive experimental verification, and as shown in Table 7, the performance improvement it brings is also significant. We believe that this correction has good implications for many methods in this field.
>
> **Q2: This paper lacks comprehensive analyses to substantiate the effectiveness of the proposed components.**
>
> **A2**: We demonstrated the effectiveness of our proposed modules or methods through ablation experiments in Table 2. Firstly, in Sec. A of appendix, we provide a detailed description of the algorithm for the label extension module, which relies on the rough action positions provided by point labels for feature upsampling and label extension. Intuitively, this will definitely lead to performance improvement. In future paper versions, we can align and visualize labels with action features to prove the accuracy of label extension in the temporal dimension. Besides, in Figure 3, we visualize the range selection of the pre-and post-optimizatized outer-inner contrast score, and it is obvious that our optimized calculation method is more accurate and can magnify the difference between the action-background segments for practical scenarios. Meanwhile, Figure 4 visualizes the comparison between our FS-PTAL and LACP in pseudo label mining methods, it is evident that the pseudo labels mined by our method are more closely aligned with GT and have a wider coverage area, indicating that our optimized method for label mining is more effective. Finally, the effectiveness of the feature separation module is also intuitive, by mapping to the feature embedding space, all action feature matrices and background feature matrices are extracted to calculate the cosine similarity and the feature separation loss, achieving action-background separation. Table 2 shows that the impact of the feature separation module on the experimental results is significant.
>
> **Q3: The paper is not self-contained in its current form, distracting the readers by making them alternate between the main paper and the appendix.**
>
> **A3**: If this paper can pass, in future versions of the paper, we will reduce the description of introduction and related work, and move the presentation of Figure 1 to the appendix. We will add more experimental tables in the main text section.
>
> **Q4: The reference format to PointTAD is wrong.**
>
> **A4**: Regarding the description of PointTAD, we do have a minor error, and we have removed its description from Table 1 and related work.

---

> > ### Comment · Reviewer_PXaQ · 2023-11-23
> > **Post-rebuttal Comment**
> >
> > I appreciate the careful response of the authors.
> >
> > Although the authors re-claim the technical novelty of the paper, it is not convincing enough to me.
> >
> > I believe some of the contributions are rather implementation tricks and the overall significance does not reach the standard bar of ICLR 2024.
> >
> > In addition, as most of the proposed modules are individually designed to improve their own counterparts, I believe there should be experiments on their superiority over the counterparts.
> >
> > However, I noticed that Table 2 only provides the results by excluding one of the modules, and the authors fail to present apple-to-apple comparisons to showcase the relative advantages of the components.
> >
> > For these reasons, I will keep my rating unchanged.

---

### Meta-Review · Area_Chair_j2BL · 2023-12-09

**Metareview:**

This paper tackles temporal action localisation from a weakly supervised perspective, using point labels (instead of the full segment labels).

All four reviews are consistent in raising concerns regarding the novelty and contribution, writing and presentation, as well as convincing experimentation.  Despite one reviewer giving a weak accept, this reviewer has concerns regarding the experimentation post-rebuttal.  The other three reviewers put forth scores ranging from reject to borderline reject).  After reading through the reviews and responses, the AC agrees with the reviewers that the paper is not yet ready for publication and recommends rejection.

**Justification For Why Not Higher Score:**

incremental contribution, writing and presentation, experimentation are problematic.

**Justification For Why Not Lower Score:**

N/A

---

### Decision · Program_Chairs · 2024-01-16

Reject